# Comparison of Dental Stone Models and Their 3D Printed Acrylic Replicas for the Accuracy and Mechanical Properties

**DOI:** 10.3390/ma13184066

**Published:** 2020-09-13

**Authors:** Marta Czajkowska, Ewa Walejewska, Łukasz Zadrożny, Monika Wieczorek, Wojciech Święszkowski, Leopold Wagner, Eitan Mijiritsky, Jarosław Markowski

**Affiliations:** 1Laryngology Department, School of Medicine in Katowice, Medical University of Silesia in Katowice, 40-027 Katowice, Poland; mrtczajkowska@gmail.com (M.C.); laryngologia@spskm.katowice.pl (J.M.); 2Department of Dental Propaedeutics and Prophylaxis, Medical University of Warsaw, 02-006 Warsaw, Poland; lwagner@wum.edu.pl; 3Faculty of Materials Science and Engineering, Warsaw University of Technology, 02-507 Warsaw, Poland; ewawalejewska@o2.pl (E.W.); monika.wieczorek.dokt@pw.edu.pl (M.W.); wojciech.swieszkowski@pw.edu.pl (W.Ś.); 4Department of Otolaryngology, Head and Neck and Maxillofacial Surgery, Tel-Aviv Sourasky Medical Center, Sackler Faculty of Medicine, Tel-Aviv 6139001, Israel; mijiritsky@bezeqint.net; 5The Maurice and Gabriela Goldschleger School of Dental Medicine, Tel-Aviv University, Tel Aviv 6997801, Israel

**Keywords:** diagnostic, 3D printing, diagnostic model

## Abstract

This study was conducted to test possibilities of application of 3D printed dental models (DMs) in terms of their accuracy and physical properties. In this work, stone models of mandibles were cast from alginate impressions of 10 patients and scanned in order to obtain 3D printed acrylic replicas. The diagnostic value was tested as matching of model scans on three levels: peak of cusps, occlusal surface, and all teeth surfaces. The mechanical properties of acrylic and stone samples, specifically the impact strength, shore D hardness, and flexural and compressive strength were investigated according to ISO standards. The matching of models’ surfaces was the highest on the level of peaks of cusps (average lack of deviations, 0.21 mm) and the lowest on the level of all teeth surfaces (average lack of deviations, 0.64 mm). Acrylic samples subjected to mechanical testing, as expected, showed higher mechanical properties as compared to the specimens made of dental stone. In the present study we demonstrated that 3D printed acrylic models could be ideal representatives in the case of use as a diagnostic tool and as a part of medical records. The acrylic samples exhibited not only higher mechanical properties, but also showed better accuracy comparing to dental stone.

## 1. Introduction

The use of dental models (DM) requires an accurate diagnosis, appropriate planning and assessment of the treatment progress in different dental specialties. As dental models are widely used in the field of dentistry, there are many guidelines which they should fulfil. It needs to be highlighted that DMs should accurately mimic the anatomical structures, which will enable their proper fixation in the articulator for occlusal analysis. Apart from that, a DM should also possess certain mechanical properties for prosthodontic and orthodontic device preparation and for their later storage as a part of medical records [1,2,3,4].

Plaster serves as a gold standard in the preparation of diagnostic models. The dental stone is classified as a third-class plaster, which has many advantages like availability and easy manufacture, leading to reduced production cost. Stone models can also accurately mimic small anatomical details of hard and soft tissue such as occlusal surface features and spacing between the tooth and the gingiva. However, dental stone, as any other available material, does also have some drawbacks. According to the literature, dental stone tends to scratch and crack, which makes it less reliable in case of device manufacturing. Once the stone model is damaged, it is near impossible to replicate it as only a single model can be cast from a single impression. In this case, this may result in permanent loss of patients’ dental information [5].

In order to overcome the problems related to dental stone models, there is also a need to develop new methods to record dental medical information. Digital models are one of the many examples which could fulfil this requirement. Digital models can facilitate the planning of the treatment by correlating occlusal conditions with patients’ own photographs [6,7]. Digital models can be fabricated with the use of 3D printers, which makes them useful in the case of device manufacturing in a dental laboratory. New types of orthodontic appliances such as aligners and some prosthodontic devices, including full and partial dentures as well as bridges, crowns on teeth, and dental implants could also be proposed and become available thanks to the development of virtual setup and 3D printing [8,9,10,11].

In the last decade, 3D printing technology has gained more interest, and not only in the field of dentistry. Some of the printing technologies dedicated to non-professional users would involve the risk of obtaining non-precise objects in the case of introduction to dentistry applications. Thus, 3SP technology (scan, spin and selectively photocured technology), developed by EnvisonTec (EnvisionTEC Inc., Dearborn, MI, USA) was implemented in this field. The 3SP technology belongs to the family of additive manufacturing, which requires the use of lasers and liquid resins to build 3D objects. The 3D models, in brief, are fabricated with 3SP printer, which consists of the laser mounted on a guide bar, the optical system, the building space, and the computer. The head with the laser is able to move low over the liquid resin, so a length of laser beam is constant. A moving platform starts from the highest level of bath with resin and moves down when each layer becomes cured. As the ability to cure very thin layers and the possibility of building stepless models are the main advantages of the 3SP technology, we decided to fabricate 3D acrylic models utilizing this method. The polymerization shrinkage is reduced by gradual curing. Models, after a process of printing, are smooth and completely cured inside. Thereafter, they should be cleaned with alcohol, dried, and exposed to ultraviolet (UV) lighting. Teeth models are made based on 3D scans. As a result of the printing process, smooth models are created [12].

In this study, the accuracy and physical properties of stone models and their 3D printed acrylic replicas were investigated. We propose that 3SP technology can be useful in fabrication of dental model substitutes and facilitate their storage as a part of medical records.

## 2. Materials and Methods

Table 1 indicates materials used in preparation of the two types of dental models for accuracy and mechanical testing.

### 2.1. Preparation of Plaster Models

The 18 g of alginate powder was mixed with 40 mL of water utilizing hand spatulation method. The impression material was then loaded onto the trays and placed in the mouths of 10 volunteers and left for gelation. Once set, the trays were removed, washed with water spray, disinfected, and dried. The casts were prepared on the same day using the type 3 stone. The stone was manually prepared in accordance to manufacturer’s instructions (100 g of powder mixed with 30 mL of water) and poured into an alginate impression. Prior to further studies, casts were removed and stored at room temperature.

### 2.2. The Diagnostic Value of Plaster and Acrylic Models

Ten stone models were scanned with the HP 3D Structured Light Scanner Pro S3 (HP Inc., Palo Alto, CA, USA, realization: Solveere, Ogrodzieniec, Poland) and digital three-dimensional models were obtained.

All digital models were used to print 3D acrylic models with the EnvisionTec ULTRA^®^ 3SP™ printer (EnvisionTEC Inc., Dearborn, MI, USA,). As the result of photopolymerization, 10 stepless models were fabricated. The supporting structures used during the printing were cut after the process of a post-curing. In order to compare diagnostic values between stone and acrylic models, the 3D printed models were also scanned with the HP 3D Structured Light Scanner Pro S3.

The maps of deviations were prepared by utilizing scans of stone models and their 3D printed copies. The maps were prepared with the HP 3D Scan Trial 5.4.0 software (HP Inc., Palo Alto, CA, USA) on 11 different levels of tolerance between 0.1 mm and 1.00 mm. The 3 levels of deviations between original models and their copies were found. The deviation map was analyzed with a tolerance of 0.01 mm for each of three detected levels.

### 2.3. Sample Preparation for Mechanical Testing

The shape and dimensions of acrylic samples for mechanical examination was designed with respect to appropriate ISO standard with the MeshLab 2016 software (ISTI-CNR, MeshLab 2016, Pisa, Italy) and printed with EnvisionTec ULTRA^®^ 3SP™ printer (EnvisionTEC Inc., Dearborn, MI, USA) utilizing modified acrylic. In order to obtain stone samples with specific dimensions, silicon forms using acrylic samples were prepared. Stone samples were cast using the blue class III stone. Prior to mechanical testing, samples were stored in a dry place at room temperature.

### 2.4. The Impact Strength Evaluation

The impact strength was determined utilizing Charpy type pendulum impact tester (Ceast, Pianezza, Italy). The study was performed with respect to PN-EN ISO 179-1:2010 [13]. In brief, seven unnotched samples with a dimension of 80 × 10 × 4 mm were clamped at one end vertically. The reduced potential energy used to fracture the samples was noted and the impact strength was calculated in accordance to Equation (1):(1)$$\mathrm{KC}=\frac{K}{S}$$
where KC is the impact strength (kJ/m^2^), K is reduced potential energy (kJ), and S is a cross-sectional area of the sample (m^2^).

### 2.5. The Shore D Hardness Test

To determine the hardness of two studied materials, analogue Durometer type D tester (Wilson Wolpert, Norwood, MA, USA) was used. Briefly, the pointed indenter penetrated the surface of the sample and the result of the hardness was noted approx. 5 s after loading. The results are presented as an average of eight measurements.

### 2.6. The Flexural Strength Evaluation

In order to determine the flexural strength, the three-point bending system (installed on MTS Q/Test) was used with respect to PN-EN ISO 178:2011 [14]. Each tested sample with the dimension of 80 × 10 × 4 mm was placed on a bending fixture consisting of two parallel supports (Figure 1). The load was then applied at the center of the specimen with cross head speed of 5 mm/min. The flexural strength was calculated using the Equation (2):(2)$$\sigma =\frac{3\mathrm{FL}}{2{\mathrm{bd}}^{2}}$$
where σ means flexural strength (MPa), F is axial load at fracture point (N), b is width of the sample (mm), and d is thickness of tested specimen (mm).

### 2.7. The Static Compression Test

The static compression test was carried out on Zwick/Roell Z250 machine (ZwickRoell, Ulm, Germany) according to PN-EN ISO 604:2006 [15]. Five specimens of each material, with the approx. dimensions of 40 mm in height and 20 mm in diameter were examined. The data obtained during this study helped calculate the compressive strength according to Equation (3):(3)$$\sigma =\frac{F}{S}$$
where σ means stress value (MPa), F is measured force (N), and S is initial cross-section of the sample (mm^2^).

### 2.8. Statistical Analysis

Data is expressed as a mean ± standard deviation (SD). An unpaired *t*-test followed by Welch’s comparison was evaluated by GraphPad Prism version 7.0 for Mac OS X (GraphPad Software, La Jolla, CA, USA), where p value 0.05 or less is considered as significant. For each mechanical test the following values were considered as significant:

0.0332 for tests of impact strength

0.0021 for tests of Shore D hardness

0.0002 for tests of compressive strength

0.0001 for test of flexural strength

## 3. Results

### 3.1. The Diagnostic Value between Stone and Acrylic Models

The map of deviations between stone and acrylic models is presented in Figure 2. The 110 deviations were compared while performing the test. The 3 levels of deviation between cast models and their 3D printed copies were detected. The first level was assigned to the lack of deviation on the peak of articulation cusps. The second level refers to the lack of deviation in articulation surfaces, whereas the third level represents the lack of deviation in all surfaces.

The matching points in area of peak cusp were determined at first. The average deviation (AD) in this level was 0.21 mm, indicating that ADs of 50% of the samples used in this study were lower than that. The highest deviation occurred in sample number two which reached the value of 0.29 mm. In the occlusal surface the highest deviation was observed at 0.56 mm, whereas the lowest was at 0.35 mm. The average level of lack of deviation on whole crown was assigned to 0.64 mm. Figure 3 shows results of this analysis.

### 3.2. Mechanical Properties

Figure 4 represents mechanical properties of two different materials tested in this study. The disintegration of stone and acrylic specimens was observed during impact strength evaluation. According to Figure 4C, the impact strength for acrylic and stone samples was 1.5 ± 0.7 kJ/m^2^ and 0.3 ± 0.1 kJ/m^2^, respectively. Shore Durometer hardness type D was also conducted, and the values of hardness were 76.0 ± 0.4 and 68.4 ± 1.4 for acrylic and stone samples, respectively (Figure 4D).

Figure 5 illustrates force/displacement curves obtained during three-point bending evaluation for two studied materials. The curves obtained for acrylic resin (Figure 5A) indicate declines in force during loading, which may indicate a gradual cracking of the material while performing the test. Figure 6B is assigned to force/displacement plots for stone samples. The material was rapidly damaged after reaching the maximum force. An unpaired *t*-test showed significantly higher values of flexural strength for acrylic samples (31.6 ± 2.5 MPa) comparing to stone ones (9.1 ± 1.3 MPa) (Figure 5B).

Figure 6 depicts force/displacement plots, which allow the calculation of the compression strength for acrylic and stone specimens. The course of plots on Figure 6A is similar to ductile materials, contrary to stone samples, where the disintegration of each sample was clearly noticed (observed as points of strength decline on Figure 6B). The compressive strength for acrylic samples was significantly higher than stone specimens, reaching approx. 93.0 ± 1.1 MPa and 17.7 ± 2.5 MPa, respectively (Figure 6A).

## 4. Discussion

Accuracy and precision in copying anatomical structures are crucial properties for achieving treatment goals. While the proper diagnosis is one of the major factors affecting the success of the treatment, there are some other essential aspects which may influence it. Specific physical properties may impact the process of fabricating prosthodontic and orthodontic appliances and so the properties of dental models should be taken into consideration. The need to deploy new precise technologies in the field of dentistry is in this case indisputable, and there are few literature references about new methods of manufacturing orthodontic devices from thermoplastics utilizing a vacuum forming method [16,17].

There are some studies which highlight the subject of precision of 3D printed dental models, but to the best to our knowledge, the great majority of them use comparison of linear measurements instead of comparing the levels of deviations. Camardella et al. note that the shape of the model base is more important in terms of accuracy and precision of 3D printed models than the chosen fabrication method. In their work, they utilized Polyjet printing and stereolithography (SLA) and three types of printed model bases (regular base, horseshoe-shaped base, and horseshoe-shaped base with a bar) as the factors influencing the destinations between reference points. The 3D printed models were scanned and the examined distance was measured on digital models. Maps of deviation were also made based on the scans. 3D printed models with a horseshoe-shaped base made with the stereolithography were characterized by significant reduction in the transversal dimension, in contrast to models made with the Polyjet technique, which were accurate independently to the shape of base [18]. A similar problem is investigated in a study by Rungrojwittayakul [19]. Contrary to the above-mentioned findings, Rebong et al. revealed that 3D printed dental models fabricated with stereolithography, Polyjet, and fused deposition modeling (FDM) statistically differ from plaster models. Since SLA and Polyjet models showed tendency toward expansion and shrinkage, the models fabricated via FDM had only minor differences and could be a good substitute for plaster models. However, FDM technology, which uses thermoplastics, is not adequate for laboratory application such as vacuum forming [20]. Another analysis of several rapid prototyping techniques in fabrication of dental models was performed by Hazevald and co-workers [21]. They conducted linear measurements on all types of printed models with a calibrated electronic digital caliper. They not only measured the width of arch, but also the mesiodistal dimension and the height of clinical crown. In agreement with the Camdarella et al. study, they highlighted that rapid prototyping techniques, such as digital light processing, Polyjet printing, and 3D printing utilizing stone powder could replace stone models in fabrication of orthodontic devices [18].

It is well-known that linear measurements are simple to achieve and that they do not require special devices or software. Moreover, detailed statistical analysis is easy to perform. Nevertheless, they do not possess certain features, which we believe are crucial in the case of our study. One of the most important points that should be highlighted is that teeth are not equally geometric lumps, and so finding the same reference points on their surfaces is complicated and the repeatability of such measurements is debatable [22]. For that reason, the obtained results could be incomplete, unreliable, and doubtful. The linear methods of comparison do not allow us to relate detailed differences between the occlusal surface of the teeth as their general shape. Although the fact that the deviations maps are a reliable form used in detection of differences between models, they still have some imperfections. The main problem occurring when using dedicated software is the presence of so-called white defects. However, their contribution could be unmentioned, as their presence is easy to estimate.

A similar method to ours, which enables the attainment of deviation maps was proposed by Favero et al. In this study the effect of printer resolution was evaluated as the main factor affecting the accuracy of orthodontic models. As in this study, in the study by Favero et al. maps of the deviation were used. Although this study concentrates on one method of 3D printing, Favero et al. describes differences between few types of additive manufacturing techniques. The results obtained due to printing 3D models with the use of different printers, shows that the type of the device can affect the printing process; however, this finding is not significant in the case of clinical use [23].

The results obtained in the present study confirm the results stated above [12,13]. The average matching on peaks of cusps is similar to the accepted not clinically significant. This small area on top of teeth is crucial in the process of proper diagnosis. There are references points which create Monson Sphere, determine the length of arch, and are useful in every measurement [24,25]. The results of our study and literature data show that 3D printing models are well matched to stone ones as far as the tops of the teeth are concerned. However, some more interesting dependences were noticed thanks to surface matching to compared models. As we can see on Figure 1, the total deviations on a whole tooth are even 6 times larger than deviation measured only on a small area on peaks of cusps. The most different areas are in grooves on occlusal surfaces and in spaces between teeth. The examination of an impact of these deviations on dental appliance accuracy requires further research.

Appropriate physical properties of material for dental models is also an important factor which could affect treatment success. However, this problem is not highlighted in the literature. The vast majority of studies on orthodontic models are related to accuracy and repeatability of models. Only one of them presents differences in dimensions stability and the resistance to compression measured for 3D printing models before and after various postprinting treatments [26]. Studies which describe diagnostic models in general in dentistry, for example in prosthodontic cases, also disregard aspects of the mechanical properties of gypsum [27,28,29]. Research by Choi et al. compares different types of replicas of stone models. For this purpose, maps of deviations were used. Investigators emphasize the advantages of this method over linear measurements. They compare traditional gypsum models, models made with 3D printing technologies, and 3D milling stone model. It is important to note that according to their results 3D milling models were fitted to original in the highest level [29].

Despite the great mechanical properties of acrylic 3D printed models, using such models are not suitable to use forever. Material used in additive manufacturing technologies are based on acrylic resins [19,30,31,32]. Such a type of material has an ability to absorb water which could impact their dimensions. Joda et al. advise to not use such models if the time of the fabrication of definitive prosthetic reconstructions is longer than 3–4 weeks [33]. However, we need to remember that there is the possibility to easily change the models if the changes of dimensions occur; every 3D printed model might be reprinted, even after a few months or years.

The results of this study show how properties of acrylic resin diverge from properties of stone. The analysis of several potential damaging factors indicates equality or advantage of the acrylic resin. Tests conducted according to ISO standards suggest that 3D printing models could reduce a risk of accidentally damaging models and deleting medical data. Results are important especially for orthodontists who lead medical practices in countries where study models must be collected.

## 5. Conclusions

In the present study we demonstrated that 3D printed dental models could be a good substitute for stone models as diagnostic tools and as a part of medical records. The modified acrylic resin has higher mechanical properties than dental stone. We assume that models made with this material could be more resistant to damage during dental appliances preparation and have more longevity when used as dental records. Nevertheless, further research is required to decide how significant the impact of 3D scanning and 3D printing is on the accuracy of dental appliances.

## Figures and Tables

**Figure 1 materials-13-04066-f001:**
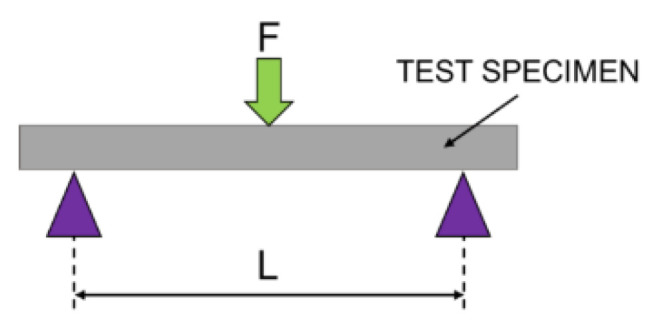
Schematic setup utilized for three-point bending test.

**Figure 2 materials-13-04066-f002:**
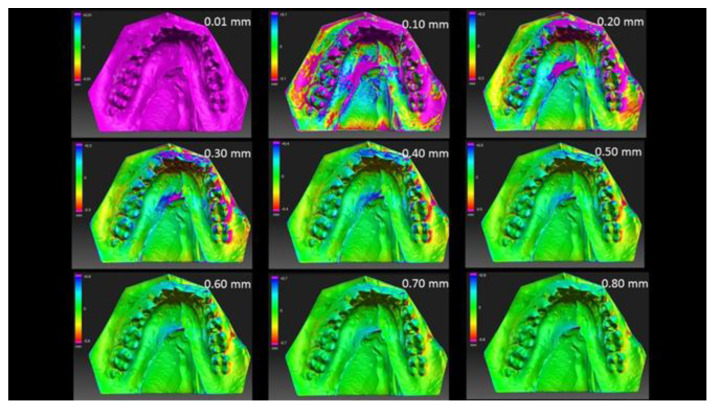
Maps of deviations prepared in accordance with the level of tolerance between 0.01 and 0.80 mm for sample number 1. Figure shows upon view areas where deviations are higher than the assumed level of tolerance as pink. Analysis was performed in three-dimensional perspective and the 0.80 mm level of tolerance was the lowest level without a deviation and are without pink areas.

**Figure 3 materials-13-04066-f003:**
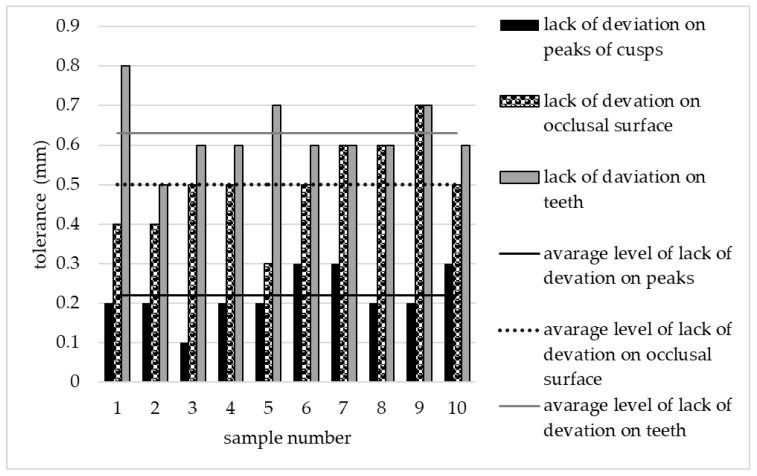
The deviation of the samples on three levels: peaks of cusps, occlusal surface, and teeth’s crowns.

**Figure 4 materials-13-04066-f004:**
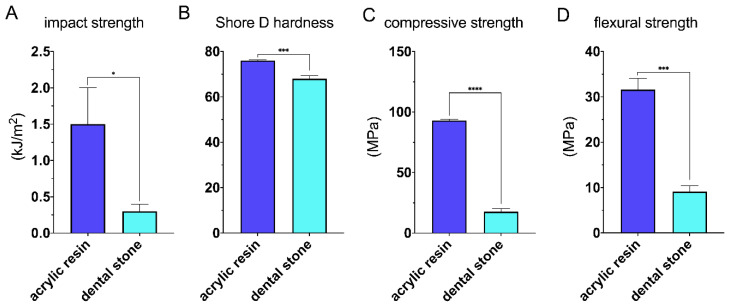
Mechanical properties of acrylic and plaster samples: (**A**) impact strength, (**B**) Shore D hardness, (**C**) compressive strength, and (**D**) flexural strength.

**Figure 5 materials-13-04066-f005:**
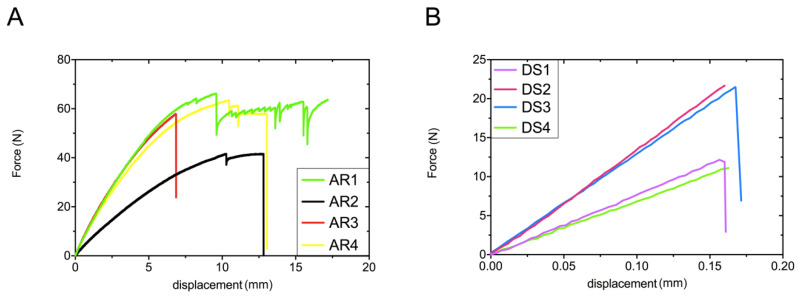
The force/displacement curves of acrylic (**A**) and stone samples (**B**) obtained during flexural strength evaluation. AR and DS means acrylic resin and dental stone, respectively (1–4 are the numbers of tested specimens).

**Figure 6 materials-13-04066-f006:**
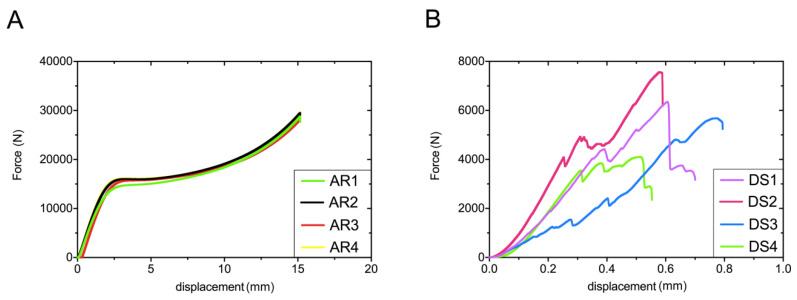
The force/displacement curves of acrylic (**A**) and stone samples (**B**) obtained during static compression test. AR and DS means acrylic resin and dental stone, respectively (1–4 are the numbers of tested specimens).

**Table 1 materials-13-04066-t001:** Materials used in the present study.

Material	Manufacturer
Chromatic alginate (class-a, type 1)	Kromopan USA Inc., Morton Grove, IL, USA
Elite HD + putty soft silicone	Zhermack SpA, Badia Polesine, Italy
Type 3 stone	Zhermack SpA, Badia Polesine, Italy
Modified acrylic resin	EnvisionTEC Inc., Dearborn, MI, USA

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
