# Peer review of "Comparison of Dental Stone Models and Their 3D Printed Acrylic Replicas for the Accuracy and Mechanical Properties"

_materials, 2020, doi:10.3390/ma13184066_

Round 1

Reviewer 1 Report

The topic of this paper is interesting and important. The methods sound. The results are meanningful and useful. There are several suggestions to improve this paper.

  1. The term m2 and kJ/m2 is not standard. 
  2. Line 144, there is a redundant space.
  3. For the literature review, the authors need to talk more about the methods used in this field. For example, finite element methods? The following papers could be refered to.

[1]  “Using the Visual Intervention Influence of Pavement Marking for Rutting Mitigation II: Visual Intervention Timing Based on the Finite Element Simulation.” International Journal of Pavement Engineering, 2019, 20 (5), 573–584.

[2] A lateral control scheme of autonomous vehicles considering pavement sustainability,Journal of Cleaner Production,Volume 256, 2020, 120669,https://doi.org/10.1016/j.jclepro.2020.120669

Reviewer 2 Report

The article fits the aims of the journal and falls within the journal purpose. The authors aim to compare usability of acrylic resin 3D printed models and dental stone models; they used a deviation map for illustrating various levels of deviations, as to evaluate the various degrees of difference between models; also assessed mechanical properties of models (compressive, flexural, impact sthrenght), establishing higher resilience to physical damage and an extended longevity for the acrylic resin model. The article is an useful contribution to the journal and the research is well conducted; however, minor changes should be taken into consideration:

Line 55 - please correct the following: “own photographs.,7”

Methodology.The methodologicaldesign is quite adequate and the techniques are well described, leading to a high level of reproductibility of the study.

Statistics. Please rephrase/ explain the thresholds for line 146: “where p value 0.0332 (*), 0.0021 (**), 0.0002 (***) and <0.0001 (****). “ for the reader could easily understand the choice of these values. If they are not chosed as a priori cut-off values and they are just the results of the statistical tests, then in the Statistics sections only the thresholds (e.g. p=0,05) should be named, and all others, if results, be moved within the results sections (and also mentioned in figures, should be the case); overall, this needs proper attention and clarifications

Discussion

Line 196 – please correct or, should be the case, elaborate on the following sentence that opens this section of the manuscript: “Authors be highlighted”

Authors would also elaborate more on similarities and differences between their article and refernce 17, namely Favero CS, English JD, Cozad BE, Wirthlin JO, Short MM, Kasper FK. Effect of print layer height and printer type on the accuracy of 3-dimensional printed orthodontic models [published correction appears in Am J Orthod Dentofacial Orthop. 2017 Dec;152(6):739]. Am J Orthod Dentofacial Orthop. 2017;152(4):557-565. doi:10.1016/j.ajodo.2017.06.012

Conclusions section

Line 266-7 -Please rephrase “We assume that models made with this material 
could be harder to damage and have more longevity. “ in order to be clearer to the reader.

Moreover, authors are advised to define all abbreviations at first appearance in text; in abstract and also elsewhere (e.g. 3SP spin…, ADs)

Grammar and punctuation must also be carefully checked within the entire article.

The article is thoroughly documented and the scientific reasoning is sound; the study is well conducted. Limitations are also mentioned.

Overall, I consider the article an useful contribution to the journal. Therefore, I recommend the manuscript for being published after suggested minor changes have been taken into consideration by the authors.
